# Population Pharmacokinetics and Dosing Regimen Optimization of Latamoxef in Chinese Children

**DOI:** 10.3390/pharmaceutics14051033

**Published:** 2022-05-11

**Authors:** Yang Wang, Dan Sun, Yan Mei, Sanlan Wu, Xinlin Li, Sichan Li, Jun Wang, Liuliu Gao, Hua Xu, Yali Tuo

**Affiliations:** 1Department of Clinical Pharmacy, Wuhan Children’s Hospital, Tongji Medical College, Huazhong University of Science and Technology, Wuhan 430016, China; wangyang@zgwhfe.com (Y.W.); meiyan@zgwhfe.com (Y.M.); lixinlin@zgwhfe.com (X.L.); lisichan@zgwhfe.com (S.L.); wangjun2@zgwhfe.com (J.W.); gaoliuliu@zgwhfe.com (L.G.); xuhua@zgwhfe.com (H.X.); 2Department of Neurology, Wuhan Children’s Hospital, Tongji Medical College, Huazhong University of Science and Technology, Wuhan 430016, China; sundan@zgwhfe.com; 3Department of Pharmacy, Union Hospital, Tongji Medical College, Huazhong University of Science and Technology, Wuhan 430022, China; wusanlan@hust.edu.cn

**Keywords:** latamoxef, epimer, population pharmacokinetics, dosing, children

## Abstract

The present study aimed to establish population pharmacokinetic models of latamoxef, as well as its R- and S-epimers, and generate findings to guide the individualized administration of latamoxef in pediatric patients. A total of 145 in-hospital children aged 0.08–10.58 years old were included in this study. Three population pharmacokinetic models of latamoxef and its R- and S-epimers were established. The stability and predictive ability of the final models were evaluated by utilizing goodness-of-fit plots, nonparametric bootstrapping, and normalized prediction distribution errors. The final model of total latamoxef was considered as a basis for the dosing regimen. A two-compartment model with first-order elimination best described the pharmacokinetics of total latamoxef. The population typical values of total latamoxef were as follows: central compartment distribution volume (V_1_) of 4.84 L, peripheral compartment distribution volume (V_2_) of 16.18 L, clearance (CL) of 1.00 L/h, and inter-compartmental clearance (Q) of 0.97 L/h. Moreover, R-epimer has a higher apparent volume of distribution and lower clearance than S-epimer. Body surface area (BSA) was identified as the most significant covariate to V, CL, and Q. Specific recommendations are given for dosage adjustment in pediatric patients based on BSA. This study highlights that a BSA-normalized dose of latamoxef was required when treating different bacteria to reach the therapeutic target more effectively.

## 1. Introduction

Latamoxef (moxalactam), a semisynthetic 1-oxo-β-lactam antibiotic, consists of R- and S-epimers, both of which are levorotatory [1]. It has an antibacterial effect that is similar to other β-lactam antibiotics, which mainly bind with penicillin-binding proteins (PBPs) to block the biosynthesis of bacterial cell walls, thus causing cell death. This drug is widely used to fight against a variety of clinically relevant Gram-positive bacteria, Gram-negative bacteria, and anaerobic bacteria [2,3]. It is one of the most commonly prescribed antibiotics against various acute and chronic bacteria infectious diseases in adults and children [4]. Although latamoxef is no longer available in some countries, including the United States, it is still used in developing countries at present. Hence, it is necessary to promote latamoxef individualized medication guidance based on pharmacokinetic (PK)/pharmacodynamic (PD) study.

Latamoxef is administered intravenously or intramuscularly in the sodium salt form. After entry into the circulation, latamoxef rapidly distributes into visceral tissues, muscles, bones, myocardium, and other tissues. Many studies have proved that latamoxef does not undergo metabolism in humans, and both epimers are usually eliminated after being unchanged by the kidney [5,6]. Moreover, latamoxef also partly undergoes biliary excretion [7].

Currently, traditional PK data on latamoxef in children population are extremely limited, especially for R- and S-epimers. In addition, latamoxef has high individual variability in PK parameters, which may affect the clinical outcomes of the patients. Nahata et al. found that the inter-individual variation in the clearance rate of latamoxef epimers is 500% in pediatric patients [8]. Additionally, the peak and trough concentrations of latamoxef showed a great difference after administration of the same dose of latamoxef in children [9]. Furthermore, despite that the efficacy of latamoxef is well established, the drug instructions lack specific dosing regimens of latamoxef for infants and children. Since PK data are limited in these patients, they are usually treated with latamoxef empirically [10]. Therefore, to optimize the therapeutic effect and reduce latamoxef resistance, therapeutic drug monitoring (TDM) is widely used to achieve the therapeutic target. However, the effects of various factors on the PK of latamoxef have little consideration in TDM.

Population pharmacokinetics (PPK) is a combination of classical PK models and population statistical models, offering a superior approach to quantitatively analyze the inter-individual and intra-individual variation of PK parameters and its influencing factors in certain populations [11,12]. Combined with Bayesian estimation method, optimal dosing regimens can be simulated to improve clinical therapeutic effects in patients.

At present, only a study conducted by Qi et al. developed a latamoxef model based on PPK, with focus on neonates and young infants [13]. Furthermore, no dose individualization of latamoxef based on PPK models has been attempted for children. Additionally, to the best of our knowledge, none of the published studies has established the PPK model of latamoxef epimers.

Based on the above considerations, the present study aimed to establish PPK models describing the PK of latamoxef and its epimers in bacterial infected children and quantitatively analyze the differences in PK parameters, thus generating findings to guide individualized administration of latamoxef in children.

## 2. Materials and Methods

### 2.1. Study Population

This study investigated patients (age ≤ 18 years) who were diagnosed with a bacterial infection and treated with latamoxef between July 2021 and November 2021 at Wuhan Children’s Hospital, Tongji Medical College, Huazhong University of Science and Technology. The exclusion criteria were as follows: participating in other trials, lack of complete dose information, intolerant to latamoxef, or other factors deemed inappropriate by the investigator for inclusion.

The study was conducted in accordance with the Declaration of Helsinki and relevant legal regulations and was approved by the Wuhan Children’s Hospital Ethics Committee for medical research ethics (number: 2021R153-E01). Written informed consent was obtained from the parents and/or guardians of the patients.

### 2.2. Dosing Regimen and Sampling

Latamoxef was administered by intravenous injection in a dose of 40–80 mg/kg/day divided into two or three doses, which would be adjusted according to the patient’s condition by referring to Chinese drug instructions. Residual blood samples were collected from routine biochemical specimens. From each patient, 1–3 samples were taken, and the patients’ dosing history and sampling time were accurately recorded. Samples were centrifuged at 1737× *g* for 5 min, and the separated serum samples were used for latamoxef concentration measurement.

Demographic and physiological characteristics of patients were acquired from the patients’ electronic medical record system, including gender, age, weight, height, white blood cell count, red blood cell count, hemoglobin concentration, hematocrit, platelets, blood urea nitrogen, serum creatinine, uric acid, cystatin C, human serum albumin, globulin, total bilirubin, direct bilirubin, alanine aminotransferase, aspartate aminotransferase, γ-glutamyl transpeptidase, high-sensitivity C-reactive protein, and procalcitonin. The estimated glomerular filtration rate (eGFR) was calculated by the modified Schwartz formula [eGFR (mL/min·1.73 m^2^) = 0.413 × Height/Serum creatinine] [14].

### 2.3. Assay of Serum Latamoxef

The latamoxef serum concentration was determined by high-performance liquid chromatography (HPLC Agilent Technologies Inc., 1260 infinity II, Santa Clara, CA, USA), followed by UV detection. The serum samples were prepared by C18 solid phase extraction columns (Cleanert ODS C18, 500 mg/3 mL, Agela Technologies, Newark, DE, USA), which were first activated with methanol and water. Next, a 0.5 mL serum sample was pipetted onto the column and then eluted with 50% methanol for analysis. The elution was carried out on a SinoChrome ODS-BP (4.6 mm × 250 mm, 10 μm) column (Elite, Dalian, China) at 30 °C. The degassed mobile phase, consisting of 0.01 M ammonium acetate solution:methanol (95:5, *v*/*v*), was pumped at 0.8 mL/min. Latamoxef was detected with UV spectroscopy at 230 nm. Chiral separation of R- and S-latamoxef was achieved by adopting this chromatographic method in the 2020 edition of the Chinese Pharmacopoeia. The method was satisfactorily linear within the range of 1.5–124 μg/mL, and the limit of quantitation was 1.5 μg/mL. The inter- and intra-day precision was less than 10%, which met the requirements of biological sample analysis.

### 2.4. PPK Modeling

PPK models were constructed by using Phoenix^®^ NLME software (Version 8.2, Pharsight Corporation, Mountain View, CA, USA). Statistical tests and graphs were performed by utilizing R software (version 3.5.1, https://www.r-project.org/, accessed on 2 July 2018, TUNA Team, Tsinghua University, Beijing, China).

The PPK model consists of structural models and random effect models. Structural models were used to describe the relationship between concentration and time, and random effect models were used to evaluate intra- and inter-individual variabilities in PPK. A two-compartment model with first-order elimination was investigated to describe the concentration–time data on the basis of a review of the literature and a visual data inspection. The choice of residual-variable model was based on changes in target function values and visualization of diagnostic plots.

The inter-individual variability was evaluated adopting an exponential model shown in Equation (1):Pi = θ* exp (ηi)(1)
where Pi is the estimated PK parameter for the individual *i*; θ is the typical value for the population parameter; and ηi is an inter-individual variation with normal distribution, mean of zero, and variance of ω^2^.

The inner-individual variability (residual variability) was described by an additive model shown in Equation (2):Y = IPRED + ε(2)
where Y is the observed value, IPRED is the individual predicted value, and ε represents the residual error of the model with a mean of zero and variance of ω^2^.

### 2.5. Covariate Analysis

The potential covariates tested were gender, age, weight, height, body surface area (BSA), white blood cell count, red blood cell count, hemoglobin concentration, hematocrit, platelets, blood urea nitrogen, serum creatinine, uric acid, cystatin C, human serum albumin, globulin, total bilirubin, direct bilirubin, alanine aminotransferase, aspartate aminotransferase, γ-glutamyl transpeptidase, high-sensitivity C-reactive protein, and procalcitonin.

The covariates were screened by stepwise forward inclusion and the backward elimination approach. In the forward inclusion, statistical significance was set to *p* < 0.05 if the objective function value (OFV) obtained was reduced more than 3.84 for 1 degree of freedom, meaning that this covariate could be put into the model. A full regression model was established by including all covariates that had a significant impact on the model. In the backward elimination, statistical significance was set to *p* < 0.01 if the OFV value was increased greater than 6.63 for 1 degree of freedom, and the covariate should be retained. The model obtained based on this is regarded as the final PPK model. The individual parameters of the final model are estimated by empirical Bayesian methods.

### 2.6. Validation of Final PPK Model

In order to assess the stability and predictive ability of the final model, several methods, including goodness-of-fit plots, nonparametric bootstrapping, and normalized prediction distribution errors (NPDEs), were employed. Goodness-of-fit plots, the most common model evaluation method, are composed of observed concentrations versus individual prediction (DV vs. IPRED), DV versus population prediction (PRED), conditional weighted residuals (CWRES) versus PRED, and CWRES versus time after dose. The stability of the model was verified by the Bootstrap method. One thousand random data sets were extracted from the original data set by resampling with the return, and the estimated median values from the 95% confidence intervals (CIs) in the bootstrapping step were compared to the estimated values of the final model parameters. The fitness of the final model to the data was further evaluated by using NPDEs. NPDEs’ results were presented in graphical form, including quantile–quantile plots, NPDE histograms, NPDE versus PRED plots, and NPDE versus time after dose plots. Assume that NPDE follows a normal distribution.

### 2.7. Simulation and Dosing Optimization

Latamoxef is a time-dependent anti-infective drug, and its PK/PD index depends on the time during which the free plasma concentration of the drug is higher than the minimum inhibitory concentration (fT > MIC) during the dosing interval. On the basis of balancing maximum efficacy with minimum toxicity and reducing drug resistance, the goal of simulation optimization is that more than 90% of patients have a free drug concentration above the MIC for a time greater than 50% of the dosing interval [15].

Based on the final model and its estimated parameter values, different dosing regimens were simulated. The Monte Carlo simulation method was used to simulate 1000 times to predict whether the latamoxef reached the therapeutic target and to provide guidelines for the individualized therapy of latamoxef in children. MIC values of 0.25, 0.5, 1, 2, 3, 4, 5, 6, 7, and 8 μg/mL were used to calculate the probability of target attainment (PTA) levels in Monte Carlo simulations.

## 3. Results

### 3.1. Study Population

A total of 165 serum concentrations of latamoxef (range 1.84–117.88 μg/mL) from 145 patients were available for PPK analysis. Of these 145 patients, 62.8% were males and 37.2% were females. The ages of patients covered a range of 0.08–10.58 years with a mean of 1.08 ± 1.63 (SD) years. Table 1 summarized the main demographic characteristics and clinical laboratory test values of the enrolled patients. According to the exclusion criteria, no patients were excluded.

### 3.2. PPK Modeling

A two-compartment model with first-order elimination was chosen to describe the PPK data of latamoxef. The central compartment distribution volume (V_1_), the peripheral compartment distribution volume (V_2_), the clearance (CL), and the inter-compartmental clearance (Q) were used as the main PK parameters.

Given that children continue to develop, ten allometric models were evaluated to study the relationship between CL and BSA (weight). As shown in Table 2, Model II was the most suitable model. Both Model III and Model IV were deserted owing to their higher OFV values. Although Model I, Model V, Model VI, Model VII, Model VIII, Model IX, and Model X had similar or lower OFV values compared to Model II, their model parameters were highly variable, and their model structure was unstable. Therefore, Model II was selected as the final model. In the selection of covariates, BSA was identified as the most important covariate of PPK models in our study. 

The final PPK model for R + S latamoxef was as follows:$$V_{1}\left(L\right)=4.84\times \left(\frac{\mathrm{BSA}}{0.39}\right)$$
$$V_{2}\left(L\right)=16.18\times \left(\frac{\mathrm{BSA}}{0.39}\right)$$
$$\mathrm{CL}\left(L/h\right)=1.00\times {\left(\frac{\mathrm{BSA}}{0.39}\right)}^{1.49}$$
$$Q\left(L/h\right)=0.97\times {\left(\frac{\mathrm{BSA}}{0.39}\right)}^{0.75}$$

In our study, r is assumed as the proportion of R-epimer in total latamoxef. Taking into consideration that the ratio of R- to S-epimer varies from 0.8 to 4.4 in drug quality standard in China (the Chinese Pharmacopoeia), the value of r is calculated accordingly, ranging from 44.44% to 58.33%. Hence, r is included in the estimated PK parameters because of the unknown proportion of each epimer from the pharmaceutical factory.

According to the above definition of r, the final model for R-epimer was as follows:$$V_{1}/r\left(L\right)=9.69\times \left(\frac{\mathrm{BSA}}{0.39}\right)$$
$$V_{2}/r\left(L\right)=33.00\times \left(\frac{\mathrm{BSA}}{0.39}\right)$$
$$\mathrm{CL}/r\left(L/h\right)=1.68\times {\left(\frac{\mathrm{BSA}}{0.39}\right)}^{1.42}$$
$$Q/r\left(L/h\right)=3.15\times {\left(\frac{\mathrm{BSA}}{0.39}\right)}^{0.75}$$

The final model for S-epimer was as follows:$$V_{1}/\left(1-r\right)\left(L\right)=8.12\times \left(\frac{\mathrm{BSA}}{0.39}\right)$$
$$V_{2}/\left(1-r\right)\left(L\right)=19.13\times \left(\frac{\mathrm{BSA}}{0.39}\right)$$
$$\mathrm{CL}/\left(1-r\right)\left(L/h\right)=2.36\times {\left(\frac{\mathrm{BSA}}{0.39}\right)}^{1.33}$$
$$Q/\left(1-r\right)\left(L/h\right)=1.89\times {\left(\frac{\mathrm{BSA}}{0.39}\right)}^{0.75}$$
where V_1_, V_2_, CL, and Q represent the individual PK parameters, and BSA is the body surface area. Table 3 showed the PPK parameter estimates of the final model. The typical values of V_1_, V_2_, CL, and Q shown in Table 3 were 4.84 L, 16.18 L, 1.00 L/h, and 0.97 L/h for latamoxef, 4.31–5.65 L, 14.67–19.25 L, 0.75–0.98 L/h, and 1.40–1.84 L/h for R-epimer; and 3.38–4.51 L, 7.97–10.63 L, 0.98–1.31 L/h, and 0.79–1.05 L/h for S-epimer.

The parameters for latamoxef are as follows: θ_V1_, typical value of central volume of distribution; θ_V2_, typical value of peripheral volume of distribution; θ_CL_, typical value of apparent clearance; θ_Q_, typical value of inter-compartment clearance; θ_1_, exponent for BSA as covariate for V_1_; θ_2_, exponent for BSA as covariate for V_2_; θ_3_, exponent for BSA as covariate for CL; θ_4_, exponent for BSA as covariate for Q; ω_V1_, square root of inter-individual variance for V_1_; ω_CL_, square root of inter-individual variance for CL; and σ, residual variability for additive error.

The parameters for R latamoxef are as follows: r, the proportion of R-epimer in total latamoxef; θ_V1/r_, typical value of central volume of distribution; θ_V1−R_, range of θ_V1/r_; θ_V2/r_, typical value of peripheral volume of distribution; θ_V2−R_, range of θ_V2/r_; θ_CL/r_, typical value of apparent clearance; θ_CL−R_, range of θ_CL/r_; θ_Q/r_, typical value of inter-compartment clearance; θ_Q−R_, range of θ_Q/r_; θ_5_, exponent for BSA as covariate for θ_V1/r_; θ_6_, exponent for BSA as covariate for V_2/r_; θ_7_, exponent for BSA as covariate for CL_R_; θ_8_, exponent for BSA as covariate for Q_R_; ω_V1/r_, square root of inter-individual variance for V_1/r_; ω_CL/r_, square root of inter-individual variance for CL_R_; and σ_R_, residual variability for additive error. 

The parameters for S latamoxef are as follows: θ_V1/1−r_, typical value of central volume of distribution; θ_V1−S_, range of θ_V1/1−r_; θ_V2/1−r_, typical value of peripheral volume of distribution; θ_V2−S_, range of θ_V2/1−r_; θ_CL/1−r_, typical value of apparent clearance; θ_CL−S_, range of θ_CL/1−r_; θ_Q/1−r_, typical value of inter-compartment clearance; θ_Q−S_, range of θ_Q/1−r_; θ_9_, exponent for BSA as covariate for θ_V1/1−r_; θ_10_, exponent for BSA as covariate for V_2/1−r_; θ_11_, exponent for BSA as covariate for CL_S_; θ_12_, exponent for BSA as covariate for Q_S_; ω_V1/1−r_, square root of inter-individual variance for V_1/1−r_; ω_CL/1−r_, square root of inter-individual variance for CL_S_; and σ_S_, residual variability for additive error.
PPK: population pharmacokinetic; SE (%), percent standard error;Bias(%) = (Median Estimate Bootstrap − Estimate Final model)/Estimate Final model × 100%.

### 3.3. Validation of Final PPK Model

The goodness-of-fit plots for the final model are shown in Figure 1. The plots show that the predicted values were extremely close to the observed values, reflecting the accuracy of the final model. The main CWRES were within ±3, indicating an acceptable fit of the model. The median parameters from the bootstrap method agreed with the respective values of the final model, with parameters falling at 95% CI, confirming the precision of the final model. The NPDE distribution and histograms for latamoxef and its epimers generally fit the theoretical N (0,1) distribution and density, as shown in Figure 2. In addition, the *t*-test, Fisher’s variance test, Shapiro–Wilks test, and Global test further proved their normal distribution, as the *p*-values exceeded 0.05 shown in Table 4. These results supported that NPDE exhibited good accuracy and stability and yielded excellent fits to predict individual and PPK parameters.

Figure 3 simulated the PK parameters obtained in this study. It depicted the predicted median of R-, S-epimer, and total latamoxef in vivo exposure over time on the assumption that patients with 0.39 m^2^ BSA received latamoxef at 250 mg, q12h continuously. R-epimer had a higher amount at steady state, owing to its slower excretion than S-epimer.

Figure 4 displayed the change of Bayesian clearance rate of latamoxef, as well as R- and S-epimer with eGFR. It could be found that the CL of latamoxef and its epimers would not change in patients with normal renal function (eGFR ≥ 60 mL/min·1.73 m^2^), suggesting that it was unnecessary to adjust latamoxef dose according to renal function for such patients in clinic.

### 3.4. Simulation and Dosing Regimen Optimization

The above results showed that CL was affected by BSA significantly. Thus, subpopulations were divided on the basis of BSA. The results of the Monte Carlo simulations are displayed in Appendix A, which shows the PTA (%) value for different latamoxef dosing regimens according to BSA when 50% fT > MIC was considered as a target attainment. In regard to achieving the goal of the PTA (%) value over 90%, Table 5 shows the dosage schedules for pediatric children at an MIC of 0.5, 1, 2, and 8 μg/mL for different BSA groups: (i) BSA of 0.2–0.4 m^2^, (ii) BSA of 0.41–0.6 m^2^, (iii) BSA of 0.61–0.8 m^2^, (iv) BSA of 0.81–1.0 m^2^, and (v) BSA of 1.01–1.2 m^2^. The results indicated that it was optimal to determine a dosing regimen of latamoxef based on BSA.

## 4. Discussion

To the best of our knowledge, this is the first study of PPK models for latamoxef epimers in pediatric children. Therefore, our study filled the gap in the PK characterization of latamoxef, as well as its epimers, and provided dose individualization reference in this population.

In the PPK analysis of latamoxef in children, we completed the establishment of the PPK models and dose optimization of latamoxef through a large single-center sample (145 patients and 165 concentrations) study. The model results showed that a two-compartment model with first-order elimination characteristics and covariate BSA could best describe the PPK characteristics of latamoxef in children. In our PPK model, the BSA covariate showed a significant correlation with the V and CL of latamoxef and its epimers. In the analysis of PK parameters in our final models, the population typical value of CL of latamoxef was 1.00 L/h, which is inconsistent with previously published CL results (0.27 L/h) obtained in neonates and young infants [13]. This discrepancy could be explained by the median age of the subjects (0.60 years; range of 0.08–10.58 years) being typically higher than that of previous studies (newborns).

Several studies have presented evidence to compare the PK parameters of R- and S-epimers by classical PK. Yamada et al. have presented data suggesting that the renal clearance of R-epimer was higher than that of S-epimer within 6 h when four American healthy volunteers received a single dose of latamoxef by intravenous injection [16]. A study of 12 American infants and children with cellulitis or epiglottitis children reported that R-epimer had higher total body clearance and apparent volume of distribution than that of S-epimer after single intravenous doses of latamoxef [8]. Apart from these findings after a single dose of latamoxef, consistent results have also been observed after injecting multiple doses of latamoxef to steady state in 30 American bacterial-infected infants and children [17]. Similar results have also been found in adults [18]. These findings provided theoretical support for the higher total body clearance and apparent volume of distribution of R latamoxef than those of S latamoxef.

Our results showing that the apparent volume of distribution of R-epimer was greater than that of S-epimer correlated well with the above studies. However, our results revealed that the CL of R-epimer was lower than that of S-epimer. The underlying reasons may be explained as follows. One can be attributed to the smaller size of samples in previous studies (≤30 American patients) than that in ours (145 Chinese patients); and the other is that racial differences (American vs Chinese) may have accounted for the different CL of latamoxef epimers. The latter needs further investigation. In addition, the difference in CL between the two isomers of latamoxef may result from the stereoselective excretion of the drug in vivo. Based on our results, the CL rate of S-epimer is 1.4 times that of R-epimer, suggesting preferential renal clearance of the S-epimer. Nevertheless, this factor has not been clearly identified, since pure R- or S-isomers are not available, and the excretion rates of each epimer cannot be precisely determined. For the reasons explained above, the present study investigated a population of Chinese children with bacterial infection, which might yield better results for predicting the PPK parameters of R- and S-epimers.

It is of interest to note that stereoisomers exist in many β-lactams, such as hydroxyl benzylpenicillin, phenethicillin, carbenicillin, and sulfocillin [19,20]. Divergent potency of stereoisomers has been found in the cases of ampicillin, hydroxyl benzylpenicillin, and sulfocillin with more active D (or R) epimer than L (or S) epimer. The possible reason may be that the more active epimer is much more similar to the pentapeptide precursor of the cell wall than the less active one is and destroys the synthesis of cell wall, thus causing bacteria death. In vitro experimentation has proved that the antibacterial activity of R latamoxef against several bacteria is twice as active as that of S latamoxef [1]. Few studies have reported PK differences between the epimers of β-lactam antibiotics, including latamoxef [21,22,23]. The exact clinical significance of the PK parameters of each isomer is not known. In our study, the CL of S-epimer was higher than that of R-epimer; this may shed light on an interesting aspect of epimeric antimicrobial therapy. However, the pharmacological activity and possible CL differences of these two latamoxef isomers require further evaluation.

Renal function plays a vital role in latamoxef excretion. Although several studies have revealed that creatinine clearance is associated with the clearance of latamoxef [5,6], BSA was identified as the only significant covariate on CL of latamoxef and its epimers in our study. The most probable reason for this was the tremendous lack of patients with moderate or severe renal insufficiency (eGFR < 60 mL/min·1.73 m^2^) in our included population.

The MICs of latamoxef are 8 μg/mL for *Pseudomonas aeruginosa*, *Staphylococcus aureus*, extended spectrum beta-lactamases (ESBLs)-producing *Escherichia coli*, ESBLs-producing *Klebsiella pneumoniae*, and ESBLs negative *Proteus*; 2 μg/mL for ESBLs-producing *Proteus*; 1 μg/mL for *Escherichia coli* and *Enterobacteriaceae*; and 0.5 μg/mL for *Klebsiella pneumoniae*, according to the data from Clinical and Laboratory Standards Institute (CLSI) and related works in the literature [2,3]. They nearly cover the common pathogenic bacteria that children are susceptible to. To achieve a cure for the different types of bacterial infection, Monte Carlo simulations have been performed with other different MIC values (MIC = 0.25/3/4/5/6/7 μg/mL); this is more suitable for clinical practice. According to a previous study, Qi et al. established a PPK model for latamoxef and drew the conclusion that an optimized dosing regimen of latamoxef is based on weight [13]. This model has provided a basis to better guide the clinical treatment of early onset neonatal sepsis [10]. However, our study has indicated that the optimized dosing regimens of latamoxef should be determined according to BSA in children, depending on the bacteria being treated. In comparison, our simulation has advantages with respect to accuracy for various bacteria.

There are some limitations in this study. First, the patient size was relatively small. Larger-scale multicenter clinical studies are needed to obtain more accurate PPK parameters. Second, the safety of simulated doses remained to be investigated in our future studies. No obvious adverse effects were observed in pediatric patients after the daily intravenous administration of 40–110 mg/kg of latamoxef in our study. Despite the limitations, the study can provide a valuable reference for personalized latamoxef treatment in children.

## 5. Conclusions

In this study, three PPK models to characterize the PK of latamoxef, as well as its epimers, were developed for the first time in Chinese children. BSA as a significant covariate was identified in the final model. This study highlights that a BSA-normalized dose of latamoxef was required for different bacterial infection in children. The PPK model should be combined with TDM to offer valuable information for optimizing dose regimens and improve antibacterial activity of latamoxef.

## Figures and Tables

**Figure 1 pharmaceutics-14-01033-f001:**
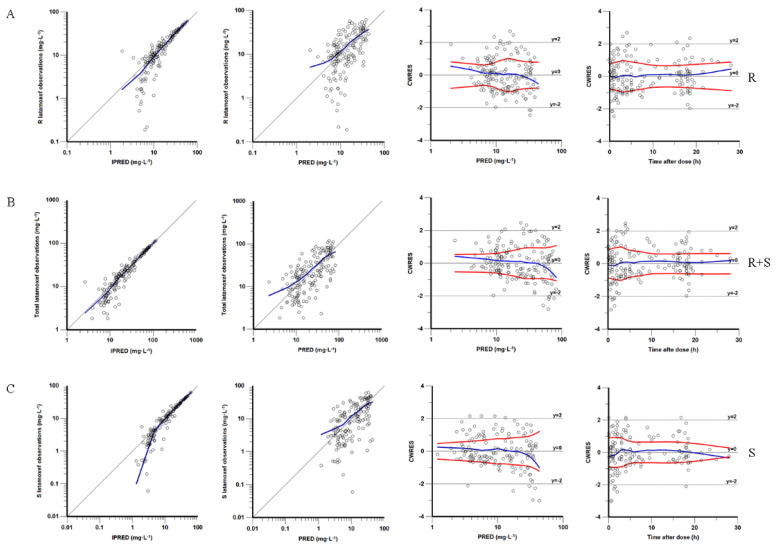
Goodness-of-fit plots of the final established population pharmacokinetic models: (**A**) R latamoxef, (**B**) R + S latamoxef, and (**C**) S latamoxef. From left to right. These plots are observed concentration versus individual-predicted concentration (IPRED), observed concentration versus population-predicted concentration (PRED), conditional weighted residuals (CWRES) versus PRED, and CWRES versus time after dose, respectively.

**Figure 2 pharmaceutics-14-01033-f002:**
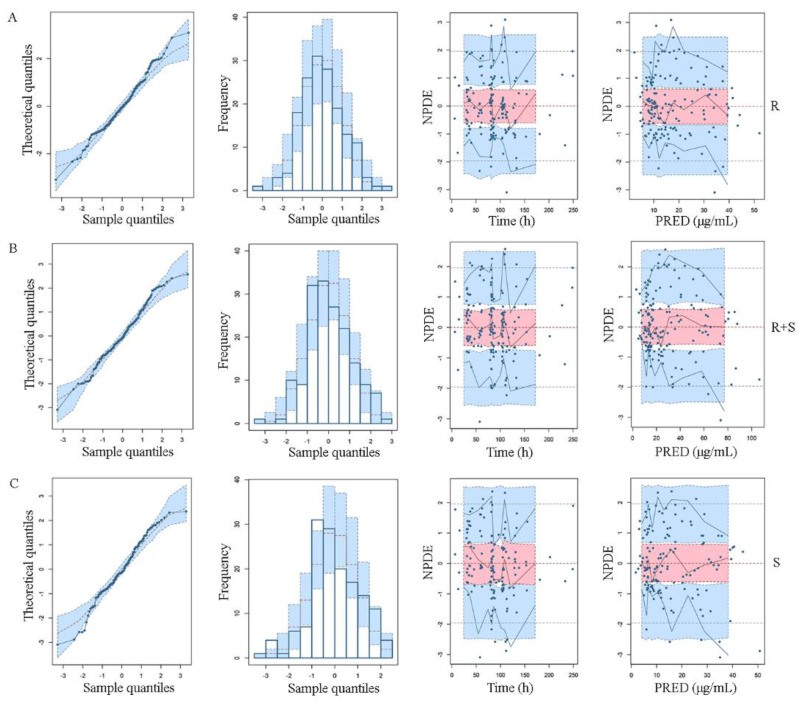
Normalized prediction distribution error (NPDE) plots for the final population pharmacokinetic models: (**A**) R latamoxef, (**B**) R + S latamoxef, and (**C**) S latamoxef. From left to right, these plots are quantile-quantile plot of NPDE versus the expected standard normal distribution, histogram of NPDE with the density of the standard normal distribution overlaid, scatterplot of NPDE versus time after dose, and scatterplot of NPDE versus population prediction (PRED), respectively.

**Figure 3 pharmaceutics-14-01033-f003:**
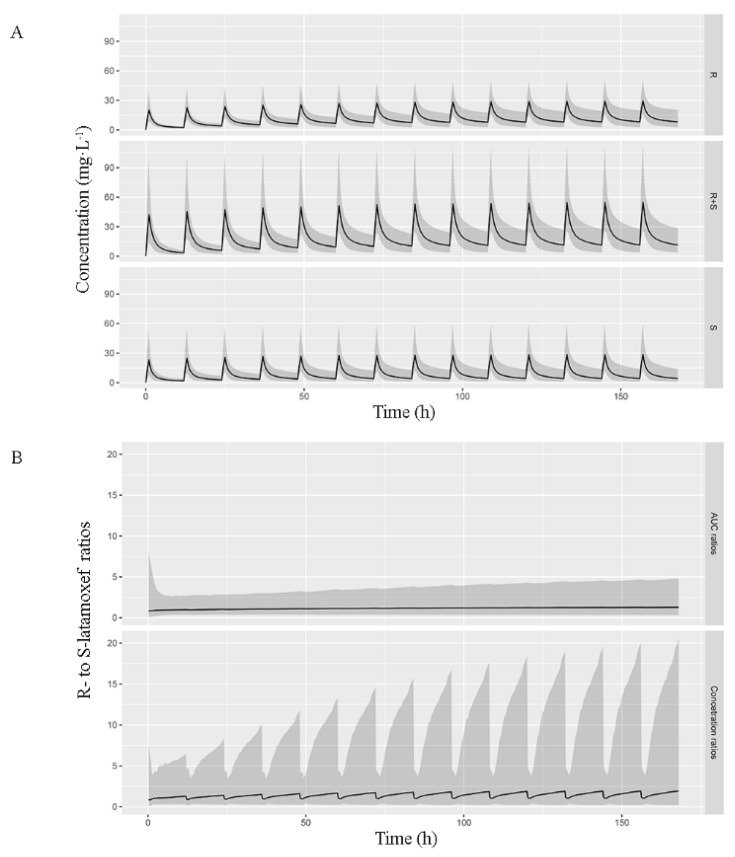
Predicted median exposure of R-, S-, and total latamoxef over time after multiple doses. (**A**) Simulation plot of predicted concentrations of R-, S-, and total latamoxef over time. The black line indicates the median, while the gray shaded area represents the 10th to 90th percentiles. (**B**) Variation interval of both AUC ratio and plasma concentration ratio of R- to S-epimer. The black line indicates the median. In the gray shaded area, the bottom represents the ratio of 10th percentile of R-epimer to 90th percentile of S-epimer, while the top represents the ratio of 90th percentile of R-epimer to 10th percentile of S-epimer.

**Figure 4 pharmaceutics-14-01033-f004:**
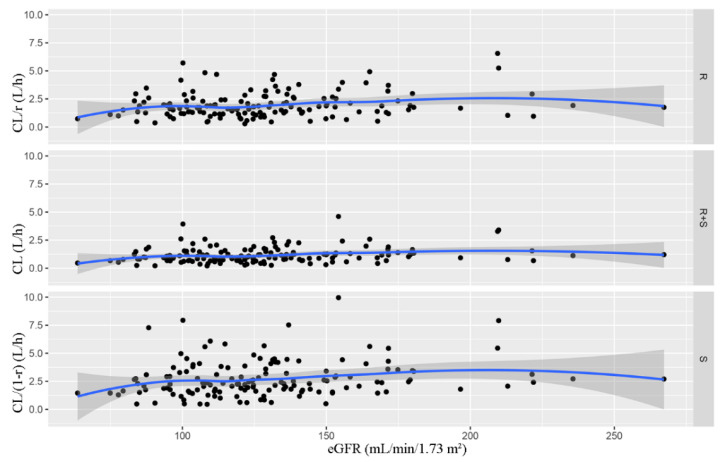
The CL of R/R + S/S latamoxef versus eGFR profile.

**Table 1 pharmaceutics-14-01033-t001:** Demographic and physiological characteristics of 145 pediatric patients.

	Number	Mean (SD)	Median (Range)
Patients	145		
Gender (M:F)	91:54		
Age (years)		1.08 (1.63)	0.60 (0.08–10.58)
Weight (kg)		8.68 (4.11)	8 (2.9–27.5)
Height (cm)		71.80 (16.98)	68.00 (49.00–140.00)
Body surface area (m^2^)		0.41 (0.14)	0.39 (0.20–1.03)
Blood urea nitrogen (mmol/L)		3.13 (1.37)	3.00 (0.60–10.10)
Serum creatinine concentration (μmol/L)		21.37 (5.97)	20.90 (9.70–48.10)
Uric acid (μmol/L)		229.18 (81.15)	224.50 (61.00–488.40)
Cystatin C (μmol/L)		1.12 (0.26)	1.08 (0.51–2.13)
Estimated glomerular filtration rate (mL/min·1.73 m^2^)		128.37 (33.79)	123.76 (63.61–267.24)
R-epimer concentration (μg/mL)		16.91 (13.94)	11.86 (0.19–60.92)
S-epimer concentration (μg/mL)		14.73 (14.52)	9.44 (0.06–62.97)
Total latamoxef concentration (μg/mL)		29.08 (26.45)	19.34 (1.84–117.88)
High sensitive C reaction protein (mg/L)		13.45 (24.54)	4.02 (0.78–156.00)
Procalcitonin (ng/mL)		0.33 (0.53)	0.14 (0.03–3.31)

SD, standard deviation. Gender (M:F): M, male; F, female.

**Table 2 pharmaceutics-14-01033-t002:** Ten candidate models of parameter estimates for clearance.

Candidate Models	Model Description			OFV
		k_1_	MF	
Model I	$\mathrm{CL}/F=\theta_{\mathrm{CL}}\times {\left(\frac{\mathrm{WT}}{{\mathrm{WT}}_{\mathrm{median}}}\right)}^{k1}\times \mathrm{MF}$	Estimated	1	1408.59
Model II	$\mathrm{CL}/F=\theta_{\mathrm{CL}}\times {\left(\frac{\mathrm{BSA}}{{\mathrm{BSA}}_{\mathrm{median}}}\right)}^{k1}\times \mathrm{MF}$	Estimated	1	1407.12
Model III	$\mathrm{CL}/F=\theta_{\mathrm{CL}}\times {\left(\frac{\mathrm{WT}}{{\mathrm{WT}}_{\mathrm{median}}}\right)}^{k1}\times \mathrm{MF}$	0.75	1	1413.9
Model IV	$\mathrm{CL}/F=\theta_{\mathrm{CL}}\times {\left(\frac{\mathrm{BSA}}{{\mathrm{BSA}}_{\mathrm{median}}}\right)}^{k1}\times \mathrm{MF}$	0.75	1	1420.83
Model V	$\mathrm{CL}/F=\theta_{\mathrm{CL}}\times {\left(\frac{\mathrm{WT}}{{\mathrm{WT}}_{\mathrm{median}}}\right)}^{k1}\times \mathrm{MF}$	0.75	$\mathrm{MF}=\frac{1}{1+{\left(\frac{\mathrm{Age}}{{\mathrm{TM}}_{50}}\right)}^{-\gamma }}$	1405.45
Model VI	$\mathrm{CL}/F=\theta_{\mathrm{CL}}\times {\left(\frac{\mathrm{BSA}}{{\mathrm{BSA}}_{\mathrm{median}}}\right)}^{k1}\times \mathrm{MF}$	0.75	$\mathrm{MF}=\frac{1}{1+{\left(\frac{\mathrm{Age}}{{\mathrm{TM}}_{50}}\right)}^{-\gamma }}$	1408.67
Model VII	$\mathrm{CL}/F=\theta_{\mathrm{CL}}\times {\left(\frac{\mathrm{WT}}{{\mathrm{WT}}_{\mathrm{median}}}\right)}^{k1}\times \mathrm{MF}$	$k_{1}=k_{0}-\frac{k_{\max}\times {\mathrm{WT}}^{\gamma }}{k_{50}^{\gamma }+{\mathrm{WT}}^{\gamma }}$	1	1408.43
Model VIII	$\mathrm{CL}/F=\theta_{\mathrm{CL}}\times {\left(\frac{\mathrm{BSA}}{{\mathrm{BSA}}_{\mathrm{median}}}\right)}^{k1}\times \mathrm{MF}$	$k_{1}=k_{0}-\frac{k_{\max}\times {\mathrm{WT}}^{\gamma }}{k_{50}^{\gamma }+{\mathrm{WT}}^{\gamma }}$	1	1406.33
Model IX	$\mathrm{CL}/F=\theta_{\mathrm{CL}}\times {\left(\frac{\mathrm{WT}}{{\mathrm{WT}}_{\mathrm{median}}}\right)}^{k1}\times \mathrm{MF}$	$k_{1}=k_{0}-\frac{k_{\max}\times {\mathrm{Age}}^{\gamma }}{k_{50}^{\gamma }+{\mathrm{Age}}^{\gamma }}$	1	1407.84
Model X	$\mathrm{CL}/F=\theta_{\mathrm{CL}}\times {\left(\frac{\mathrm{BSA}}{{\mathrm{BSA}}_{\mathrm{median}}}\right)}^{k1}\times \mathrm{MF}$	$k_{1}=k_{0}-\frac{k_{\max}\times {\mathrm{Age}}^{\gamma }}{k_{50}^{\gamma }+{\mathrm{Age}}^{\gamma }}$	1	1406.05

OFV, objective function value; MF, factor for maturation; θ_CL_, typical value of clearance; WT, weight; BSA, body surface area; TM_50_, maturation half-time; γ, Hill coefficient defining the steepness of the sigmoidal curve; k_1_, allometric exponent; k_0_, the exponent at a theoretical weight of 0 kg or age at 0 years; k_max_, a maximum decrease of the exponent; k_50_, the weight or age when a 50% drop in the maximum decrease of the exponent is achieved.

**Table 3 pharmaceutics-14-01033-t003:** Latamoxef and its R- and S-epimers PPK parameter estimates of the final model and bootstrap validation.

Group	Parameter	Final Model		Bootstrap Analysis			Bias (%)
		Estimate	SE (%)	2.5th Percentile	Median Estimate	97.5th Percentile	
R + S	θ_V1_ (L)	4.84	15.85	3.30	4.66	6.53	−3.72
	θ_V2_ (L)	16.18	47.35	9.05	16.54	26.41	2.22
	θ_CL_ (L/h)	1.00	9.05	0.82	0.99	1.15	−1.00
	θ_Q_ (L/h)	0.97	15.93	0.71	0.98	1.62	1.03
	θ_1_	1.00 (fixed)					
	θ_2_	1.00 (fixed)					
	θ_3_	1.49	14.69	1.05	1.46	1.92	−2.01
	θ_4_	0.75 (fixed)					
	Inter-individual						
	ω_V1_ (%)	105.04	29.96	26.70	110.78	194.86	5.46
	ω_CL_ (%)	28.84	31.45	11.26	28.08	44.90	−2.64
	Residual variability						
	σ (mg/L)	7.29	11.49	5.06	7.07	9.05	−3.02
R	θ_V1/r_ (L)	9.69	16.00	6.69	9.51	13.35	−1.86
	θ_V1−R_ (L)	4.31–5.65					
	θ_V2/r_ (L)	33.00	33.75	21.54	33.23	48.37	0.70
	θ_V2−R_ (L)	14.67–19.25					
	θ_CL/r_ (L/h)	1.68	9.71	1.37	1.67	1.93	−0.60
	θ_CL−R_ (L/h)	0.75–0.98					
	θ_Q/r_ (L/h)	3.15	21.70	2.08	3.14	5.24	−0.32
	θ_Q−R_ (L/h)	1.40–1.84					
	θ_5_	1.00 (fixed)					
	θ_6_	1.00 (fixed)					
	θ_7_	1.42	18.07	0.97	1.42	1.89	0.00
	θ_8_	0.75 (fixed)					
	Inter-individual						
	ω_V1/r_ (%)	65.11	41.41	7.03	67.79	128.55	4.12
	ω_CL/r_ (%)	35.37	34.46	15.80	34.95	54.10	−1.19
	Residual variability						
	σ_R_ (mg/L)	5.33	13.07	3.91	5.21	6.27	−2.25
S	θ_V1/(1−r)_ (L)	8.12	18.13	5.45	8.55	12.68	5.30
	θ_V1−S_ (L)	3.38–4.51					
	θ_V2/(1−r)_ (L)	19.13	48.39	9.50	17.86	58.69	−6.64
	θ_V2−S_ (L)	7.97–10.63					
	θ_CL/(1−r)_ (L/h)	2.36	9.65	1.81	2.32	2.85	−1.69
	θ_CL−S_ (L/h)	0.98–1.31					
	θ_Q/(1−r)_ (L/h)	1.89	20.26	0.93	1.81	3.46	−4.23
	θ_Q−S_ (L/h)	0.79–1.05					
	θ_9_	1.00 (fixed)					
	θ_10_	1.00 (fixed)					
	θ_11_	1.33	20.48	0.78	1.34	1.98	0.75
	θ_12_	0.75 (fixed)					
	Inter-individual						
	ω_V1/(1−r)_ (%)	116.20	33.21	25.41	129.09	232.77	11.09
	ω_CL/(1−r)_ (%)	43.20	28.80	0.90	40.04	79.18	−7.31
	Residual variability						
	σ_S_ (mg/L)	3.81	9.24	1.79	3.72	5.27	−2.36

**Table 4 pharmaceutics-14-01033-t004:** NPDE results of the final model.

Item		Model	
	R + S	R	S
NPDE mean (SE)	0.04 (0.08)	0.03 (0.09)	−0.01 (0.09)
Variance (SE)	1.10 (0.12)	1.14 (0.13)	1.20 (0.14)
Skewness Value	0.06	0.24	−0.15
Kurtosis Value	−0.05	0.15	−0.01
*t*-test *p*-value	0.626	0.722	0.937
Fisher variance test *p*-value	0.349	0.214	0.100
Shapiro–Wilks test of normality *p*-value	0.367	0.204	0.119
Global adjusted *p*-value	1.000	0.613	0.301

NPDE, normalized prediction distribution errors; SE, standard error.

**Table 5 pharmaceutics-14-01033-t005:** Dosing regimen of latamoxef by Monte Carol simulations.

BSA Group	MIC_90_ (μg/mL)			
	0.5	1	2	8
0.2–0.4 m^2^	50 mg, q12h	100 mg, q12h	150 mg, q12h	200 mg, q6h
0.41–0.6 m^2^	100 mg, q12h	200 mg, q12h	375 mg, q12h	475 mg, q6h
0.61–0.8 m^2^	200 mg, q12h	375 mg, q12h	400 mg, q8h	625 mg, q6h(2h)
0.81–1.0 m^2^	300 mg, q12h	550 mg, q12h	600 mg, q8h	950 mg, q6h(2h)
1.01–1.2 m^2^	500 mg, q12h	925 mg, q12h	900 mg, q8h	1400 mg, q6h(2h)

BSA, body surface area; MIC, minimum inhibitory concentration.

## Data Availability

The data presented in this study are available on request from the corresponding author. The data are not publicly available due to ethical reasons as per local guidelines.

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
