# Peer review of "Population Pharmacokinetics and Dosing Regimen Optimization of Latamoxef in Chinese Children"

_pharmaceutics, 2022, doi:10.3390/pharmaceutics14051033_

Round 1

Reviewer 1 Report

The manuscript deals with the population pharmacokinetics and dosing regimen optimization of latamoxef in Chinese children. The study is well conducted and analyzed. I recommend its publication in Pharmaceutics.

Reviewer 2 Report

This paper describes dosing regimen optimization of Latamoxef, an antibiotic used to treat various Gram-positive, Gram-negative and anaerobic bacteria, in Chinese children and uses Population pharmacokinetics to determine this. The methods used in this study are robust and sound and I believe the conclusion drawn are informative in aiding dosing of the drug in children. However, I believe there are a number of points that need to be addressed and these are summarized below: 

Specific Comments

- Why was the commonly used 100mg/kg loading dose not given? 

- Cite the sentence in line 56 about patients being treated with latamoxef empirically. 

- Need to be more specific and clearer in certain parts. In the abstract, you claim the study population ranges between 0.08-10.58 years old (line 20), but then in line 77 say <18 years.  Why use 18 years and not 11 years here. Just doesn’t read very well. 

- Why was the dose range of 40-80 mg/kg selected? Please provide justification for this?

- Can you clarify if line 363 means that racial differences were not taken into account in previous studies

- Sentence in line 376 and 377 is confusing. Why would the R epimer work in that way and the S epimer not?

- Figure 1, DV vs IPRED and DV vs PRED…since you’re using an additive residual error model, which may behave differently at lower concentrations vs higher concentrations, please log-transform both axes to better depict the ability of this additive RUV model at both the low and high concentration ranges.

- Figure 3, since you simulated 1000 monte carlo replicates, please depict both the median (as a bold line) and 80% or 90% prediction interval (as a shaded region) so the reader can better understand the predicted variability of the model.  Further, while simulating the R/S ratio is informative, so too would have been depicting the predicted concentrations of R, S and R+S over time from multiple injections, depicted as median +/- prediction interval.

- Discussion, pg 12, lines 365-368; glomerular filtration is more often than not a size-exclusion function, but with additional input into the renal tubule from active transport.  Please specify if R vs S-epimers are suspected to have differential binding to active transporters in the kidney, or if there is literature evidence of differential, stereoselective renal clearance of a particular chiral molecule.  Conversely, the lower clearance of R-epimer may simply be due to the fact that R-epimer is more active and is staying at the site of action in bacteria (i.e. a separate clearance function that cannot be accounted for since upon sample analysis, R-epimer “sequestered” or cleared by the bacteria site of action would still be present upon extraction of the blood sample for LC-UV analysis.

- Additionally, there are a number of grammatical errors throughout the text:

            - In the abstract, line 24 has “was” and this can be removed from the sentence. 

            - The phrase in line 47 needs to be rephrased e,g, “Biliary excretion of latamoxef has also             been observed”.

            - Line 49. Exchange “Meanwhile” with “In addition”

            - Line 54 should be rephrased

            - Lines 400 and 401 can be rephrased.

Reviewer 3 Report

The authors studied the population PK in children for the stereoisomer of latamoxef, a third-generation cephalosporin. I have some comments that I believe might help the authors in increasing the impact of this manuscript.

  1. Line 40-42: It is difficult to agree with the contents of this part.
  • Latamoxef is no longer available in some countries, including the United States, because of the sometimes fatal adverse reactions (prolonged bleeding time and several cases of coagulopathy). Although it is relative, can it be expressed as a safe drug? I think more explanation is needed.
  • There seems to be a problem with it being one of the most commonly prescribed antibiotics.
  1. A recent paper reported in Qi et al. is missing. Authors have to cite and discuss them in this article.  Qi H, Wu YE, Liu YL, Kou C, Wang ZM, Peng XX, Chen L, Cui H, Wang YJ, Li JQ, Zhao W, Shen AD. Latamoxef for Neonates With Early-Onset Neonatal Sepsis: A Study Protocol for a Randomized Controlled Trial. Front Pharmacol. 2021;12:635517
  1. It may be difficult due to the nature of the field, but some references are too old. Please update to the latest one.
  2. Check an abbreviation. Please consistent use of abbreviations throughout the manuscript. A few examples,
  • Line 28, 301: Change “Body surface area” to “BSA”.
  • Please check “population pharmacokinetics”.
  • Line 226: Change from “PoPPK” to “PPK”.
  1. Please check blank between number and unit.
  2. Materials and Methods: The name of manufacturer should be specified with city, state (for USA) and country.
  3. Please unify volume unit. e.g., mL vs ml.
  4. Line 66: Change from “Hui Qi” to “Qi et al.”
  5. Line 347: Change form “H Yamada” to “Yamada”.
  6. Line 131: What means “ith individual”?
  7. References: Please check reference style. In particular, Author indications, journal abbreviation names, etc.

Reviewer 4 Report

Latamoxef PK data in children is limited, and this study provides important information.

Line 30: What does "BAS" mean?

Line 90: "Samples were centrifuged at 15,000 rpm/min" may be inappropriate. The term "rpm" means "rotation per minute". Additionally, "rpm" should be converted to "x g".

Line 95: "blood cell counts"?

Lines 96-100/ 140-145:  Some examination items were repeated. Please confirm this sentence carefully.

The author should describe the methods for "Chiral separation" for R and S.

Results: Were all laboratory data obtained at the same time in each patient?

(e. g.Was Cys-C routinely obtained in your institution??)

Line 101: Are there any patients (neonates) who should correct the k value (0.413), such as preterm neonates.

Line 110: Please check carefully. No ")" was found.

This study did not describe the true PK parameter for R- and S-epimer because the assuming R/S ratio was used. No discussion and analysis about R- and S-epimer are required. The doses of R- and S- LMOX may not truly reflect the R- and S-LMOX concentration in the human body.

Figure 4: Why did the author describe the CL for both or each epimer. In fact, we did not face the situation in only one epimer exists in the human body.

Discussion was well described

Round 2

Reviewer 4 Report

rpm/min may be inappropriate.

Please delete "/min".

No other revision may be required.
